# Combined Modeling Study of the Binding Characteristics of Natural Compounds, Derived from Psoralea Fruits, to β-Amyloid Peptide Monomer

**DOI:** 10.3390/ijms23073546

**Published:** 2022-03-24

**Authors:** Awwad Radwan, Fars Alanazi

**Affiliations:** 1Kayyali Chair for Pharmaceutical Industry, Department of Pharmaceutics, College of Pharmacy, King Saud University, P.O. Box 2457, Riyadh 11451, Saudi Arabia; 2Department of Pharmaceutical Organic Chemistry, Faculty of Pharmacy, Assiut University, Assiut 71526, Egypt; afars@ksu.edu.sa

**Keywords:** β-amyloid protein, Alzheimer’s disease, docking, MD simulations

## Abstract

A dysfunctional protein aggregation in the nervous system can lead to several neurodegenerative disorders that result in intracellular inclusions or extracellular aggregates. An early critical event within the pathogenesis of Alzheimer’s disease is the accumulation of amyloid beta peptide within the brain. Natural compounds isolated from Psoralea Fructus (PF) have significant anti-Alzheimer effects as strong inhibitors of Aβ42 aggregation. Computer simulations provide a powerful means of linking experimental findings to nanoscale molecular events. As part of this research four prenylated compounds, the active ingredients of Psoralea Fructus (PF), were studied as Aβ42 accumulation inhibitors using molecular simulations modeling. In order to resolve the binding modes of the ligands and identify the main interactions of Aβ42 residues, we performed a 100 ns molecular dynamics simulation and binding free energy calculations starting from the model of the compounds obtained from the docking study. This study was able to pinpoint the key amino acid residues in the Aβ42 active site and provide useful information that could benefit the development of new Aβ42 accumulation inhibitors.

## 1. Introduction

Alzheimer’s, Alois Alzheimer’s disease (AD), is a type of dementia that affects memory, language, and behavior [1]. According to WHO forecasts, its global prevalence will double over the next four decades, reaching 114 million patients by 2050 [2]. In addition to having a significant social impact, this would undoubtedly raise the financial burden on healthcare systems around the world [3]. Dementia affects 46.8 million people globally, with dementia care costing USD 818 billion in 2010 [4]. By 2030, it is expected that 74.7 million people will have dementia, with the expense of caring for them reaching USD 2 trillion. Alzheimer’s disease causes the patient’s functionality to deteriorate over time, leading to insubstantial and long-term incapacity 7 to 10 years after diagnosis, and eventually death. When delirium lasts for at least 6 months without additional symptoms, the diagnosis of Alzheimer’s disease is highly likely [5]. Alzheimer’s disease (AD) is classified according to its onset age and whether it was acquired naturally or as a result of genetic alterations. Familial Alzheimer’s disease (FAD) is a genetically influenced, early-onset (around 40 years of age) disease that accounts for about 2% of all cases of Alzheimer’s disease. The most common type of Alzheimer’s disease is sporadic AD, which is further classified into early-onset and late-onset variants. The early-onset form is identified in people younger than 65 years old (3–5% prevalence), while the late-onset type is diagnosed in people older than 65 years old (95–97% prevalence). Alzheimer’s disease is a multifaceted illness that is influenced by a variety of circumstances. The full pathophysiology of AD is still unknown due to the complexity of human brains, as well as a lack of appropriate animal models and research tools. There have been numerous hypotheses created about Alzheimer’s disease, including β-amyloid (Aβ), Tau, cholinergic neuron damage and oxidative stress, inflammation, and so on. As a result, a lot of work has gone into developing anti-AD medications based on these theories.

The amyloid theory stands for the most likely hypotheses proposed to account for the cause of AD [5]. Aβ40 peptides are the major constituents of AD-associated amyloid plaques while Aβ42 peptides are the most toxic species [6]. Aβ oligomer poisonous effects have several mechanisms including adsorption, insertion, aggregation, and pore formation in the membrane. Furthermore, Aβ oligomers can be harmful due to interactions with receptors in the membrane and oxidative stress. Figure 1 shows the pleiotropic effects of Aβ peptides. Amyloid fibril development is a multi-state process that begins with the cleavage of amyloid fragments from the transmembrane APP, followed by the misfolding of Aβ monomers, which result in a variety of forms including unfolded clusters, beta-sheet oligomers, bigger fibrils, and amyloid plaques. These amyloid aggregates may interact strongly with the membrane because of their near proximity to cell surfaces.

The schematic representation of the monomer within Aβ40 (major and less neurotoxic) and Aβ42 (minor and highly neurotoxic) fibrils is shown in Figure 2. In addition to hydrophobic interactions, a salt bridge links Asp23 and Lys28 to stabilize the turn conformation. Within Aβ40, the residues 1–10 are unstructured while the residues 11–40 are structured as β-turn-β folds [7,8]. Hydrophobic interactions between Phe19 and Ile32/Leu34/Val36, His13 and Val40, and Gln15 and Val36 result in side chain compacting. Residues 1–17 in Aβ42 fibrils are unfolded, while residues 18–42 are β-turn folded and there are contacts between Phe19 and Gly38 [9] and between Met35 and Ala42. The salt bridge between Asp23 and Lys28 and hydrophobic interactions stabilize the turn structure of both Aβ40 and Aβ42 [10].

Currently, a few useful techniques to decrease Aβ-induced harm are being offered, such as β- and γ-secretase inhibitors, which reduce the synthesis of Aβ peptides. The toxicity of Aβ oligomers can also be reduced by employing aggregation inhibitors or pore/channel blockers [11]. A seven-membered macrocyclic molecule has demonstrated promising inhibition of amyloid fibrillation by holding aromatic residues such as phenylalanine (Phe) inside its hydrophobic cavity and hosting the charged or polar residues on its surface. Definitely, Phe residues initiate the hydrophobic clustering of amyloid proteins and is considered central in the Aβ fibrillation process [12]. Shuaib and Goyal observed that a sulfonamide molecule stabilizes native α-helix conformation over the β-sheet form using 200 ns MD simulation [13]. Moreover, from molecular dynamic (MD) simulations it was reported that flavonoid compounds (E)-5-(4-hydroxystyryl)quinolone-8-ol, J147 derivative, and edaravone prevent the amyloid-Aβ42 conformational transition by disrupting its Asp23–Lys28 salt bridge [14,15].

The majority of medications are most effective in persons who are in the early or middle stages of Alzheimer’s disease. It is crucial to note, however, that none of the current drugs can cure Alzheimer’s disease. Treating the symptoms of mild to moderate Alzheimer’s disease can provide people more comfort, dignity, and independence while also encouraging and assisting their caretakers. Cholinesterase inhibitors such as galantamine, rivastigmine, and donepezil are given for mild to moderate Alzheimer’s symptoms. Some cognitive and behavioral symptoms may be reduced or controlled with the use of these medicines [16]. Some people with moderate to severe AD may be able to keep certain everyday functions for a bit longer with medication than they would without it. Memantine, an N-methyl D-aspartate (NMDA) antagonist, for example, may assist a person with Alzheimer’s disease in preserving his or her capacity to use the restroom independently for several months longer, which is beneficial to both the person with Alzheimer’s and carers. Memantine is thought to act by modulating glutamate, a key neurotransmitter [17]. When glutamate is produced in excess, it can cause brain cell death. Because NMDA antagonists and cholinesterase inhibitors function in various ways, they can be administered together. Furthermore, the FDA has approved donepezil, the rivastigmine patch, and a memantine and donepezil combination medicine for the symptomatic treatment of moderate to severe Alzheimer’s disease [18].

Disease-modifying treatments reduce Aß synthesis, improve Aß clearance, and prevent Aß aggregation into amyloid plaques. Immunotherapy has also piqued interest since it aims to eliminate Aß peptides, which can affect cognitive decline directly or indirectly. The only disease-modifying medicine currently approved to treat AD is aducanumab. This drug is an immunotherapy that targets the protein β-amyloid and helps to eliminate amyloid plaques, which are brain lesions linked to AD [19]. Aducanumab is a recombinant human IgG1 antibody that has a >10,000-fold selectivity for soluble Aβ aggregates and insoluble fibrils over monomers. It detects the Aβ sequence’s 3–7 amino-terminal residues [20]. Aducanumab is a monoclonal or polyclonal humanized anti-Aβ antibody that is used as a passive anti-Aβ immunotherapy. Solanezumab, gantenerumab, crenezumab, and BAN2401 are four more antibodies now being studied in patients with early Alzheimer’s disease, persons with familial AD in the preclinical stage, and asymptomatic adults at high risk of developing the disease. Anti-amyloid-Aβ immunotherapies that are active were later developed. The delivery of an Aβ antigen that can elicit an immune response to Aβ is known as active immunization. For example, AN-1792, a pre-aggregated Aβ1-42 antigen and vanutide, numerous short Aβ fragments, reduced Aβ deposits in the brains of Alzheimer’s patients but had no cognitive or clinical advantages [21]. Only CAD106, an active anti-Aβ vaccine, is now being tested in phase III studies. The Aβ antigen CAD106 is made up of numerous copies of the Aβ1–6 segment. Antibodies induced by CAD106 interacted with Aβ monomers and oligomers in cell cells, blocking A toxicity. However, no research on the behavioral or cognitive effects of CAD106 in animal models of AD is available [22]. BACE (beta-secretase) is one of the enzymes that clips APP and allows β-amyloid to develop. Interrupting this process may lessen the amount of β-amyloid in the brain and, in turn, delay the onset of AD. Clinical studies for CNP520, a BACE1 inhibitor that prevents the BACE1 enzyme from chopping up APP, are expected to be completed by 2025.

Interconnections between cholinergic abnormalities and other pathophysiological features of AD, such as abnormal A and tau cascade, point to the development of novel multi-drug ligands (MTDL), as the pharmacophoric functions responsible for AChE inhibition could be combined with pharmacophoric fragments thought to interact with BACE-1. AChE, BACE-1, GSK-3, monoamine oxidase, metal ions, and even A aggregation have all been studied as AD targets. The most often used technique involves combining two or more molecular scaffolds with known features or targets into a single molecular entity. Despite their preclinical success, no reports of any of these drugs have been published. Clinical trials, and more research is needed to address some of their shortcomings [23].

Psoralea Fructus (PF), dried mature fruits (Psoralea Fructus, PF) of the Leguminosae plant, Cullen corylifolium (syn. Psoralea corylifolia), is distributed in India, China, and Southeastern Asian countries. PF has been utilized as a conventional Chinese pharmaceutical for well-being supplement fixing [24]. PF has been reported to contain compounds with molecular diversity, with a few of these displaying estrogen-like, anti-oxidant, osteoblastic, anticancer, antidepressant, anti-inflammatory, hepatoprotective [25], and antimicrobial activities [26]. Furthermore, in an AD mouse model, long-term dietary intakes of PF’s total prenylflavonoids (TPFB) at 50 mg/kg day significantly enhanced cognitive function and AD-like neurobiochemical alterations [27], and could modulate amyloid β-peptide 42 (Aβ42) aggregations in vitro [28]. Four flavonoid compounds (**1**–**4**) of PF were suggested to generate valuable effects in AD prophylaxis and treatment. The inhibitory rates percent of 100 µM PF compounds **1**–**4** on Aβ42 aggregation were reported as 98, 90, 68, and 19%, respectively [29]. The goal of the study was to model the mechanism of binding of Aβ42 protein to Aβ42-aggregation inhibitors 1–4 and identify the binding site residues that are important for action. Molecular docking, molecular dynamics (MD) simulations, and molecular mechanics/Poisson–Boltzmann surface area (MM/PBSA) computations are also part of the research. The docking results can help to understand the binding process. To investigate the role of each active site residue in inhibiting Aβ42 accumulations, the researchers performed energy calculations and energy per-residue decomposition analyses. Furthermore, the results can provide structural insights to design more active compounds as novel Aβ42 aggregation inhibitors.

## 2. Results and Discussions

### 2.1. Docking Study

Autodock 4.0 [30] was used to obtain understanding of the probable binding modes of compounds **1**–**4** (Figure 3). To incorporate all probable binding locations, a big box was defined. As a result, the docked ligands adopted the most favorable binding poses and were ranked based on their docking scores. A total of 100 docked conformations were produced and clustered and the docked conformation was selected depending on binding energy, hydrophobic interaction, and hydrogen bonding. The selected binding poses of compounds **1**–**4** exhibited a binding energy of −5.23, −4.78, −4.27, and −4.00 kcal/mol, respectively. The binding poses of the binders were laid inside a binding site delineated with the side chains of residues Glu11, His14, Gln15, Val18, Phe19, Phe29, Glu22, Asp23, and Asn27 (Figure 4 and Table 1). The docked ligand–protein complex was used as the initial structure for the MD simulations of each of the compounds **1**–**4**.

### 2.2. Molecular Dynamics Simulations

Because the α-helix is the primary shape of Aβ42 in the membrane, we picked the amyloid Aβ42 conformation (PDB ID 1Z0Q), which corresponds to the traditional α-helical conformation of Aβ42. Aβ42, on the other hand, can take on several conformations in solution, including random coil, β-strand structure, and stable turns and bends. Furthermore, various structural changes have been observed as occurring during aggregation. During Aβ folding and assembly, both the random coil to β-sheet and α -helix to β-sheet transitions occurred. The α-helix to β-strand transitions are particularly important in the fibril formation process [31]. The MD simulation methods are widely used to study the conformation changes in protein–ligand interactions [32,33,34]. MD simulations were thought important to gain insight into ligand binding modes derived through molecular docking due to the flexibility of the protein structure and its electrostatic interactions with the ligand [35].

The aggregation of Aβ42 peptide monomers into oligomers, protofibrils, and finally fibrils is the first step in the creation of neurotoxic fibrils. The β-turn folding of the amyloid peptide monomers is required for the aggregation of Aβ42 peptides. The β-turn conformation is stabilized by a salt bridge link between Asp23 and Lys28, as well as hydrophobic contacts between Phe19 and Ile32/Leu34/Val36, His13 and Val40, and Gln15 and Val36 (Figure 2) [7,8,9,10]. Disturbances in these interactions destabilize the β-turn conformations, release the side chain compactness, and ultimately impede Aβ42 aggregation and prevent the production of A42 fibrils. The simulation of Aβ42 in the presence of compounds **1**–**4** is described here, with a focus on the interaction of these inhibitors with amino acid residues required for β-turn folding. AMBER 18 software was used to perform MD simulations on the Aβ peptides in order to achieve this goal. These chemicals are thought to change the structure of a protein’s hydrogen bond network in water, reducing intermolecular hydrogen bonds, electrostatic interactions, and hydrophobic interactions. In the presence of each inhibitor, information about structural features such as intermolecular hydrogen bonding (HB) and root-mean-square derivations (RMSD) were acquired and averaged. The complexes’ binding free energy GMM/GBSA also aided in understanding the inhibitor–protein interaction mechanism.

The conformational changes of compounds **1**–**4** were observed starting from the initial docked minimized complex and proceeded for over 100 ns. The MD analyses on potential energy, temperature, and pressure for the system are given in Appendix A. The difference between the protein–ligand coordinates’ initial structural conformation and its final position was measured using root mean square deviation (RMSD). The deviations made during the simulation can be used to measure the system’s stability in relation to its starting conformation. The fewer the deviations, the more stable the structure. The RMSD value for the atom coordinates of Aβ42-binder complexes was computed for a 100 ns simulation to ensure that the systems were stable (Figure 5a–d). The Aβ42–compound **1** complex system equilibrated after 52 ns, according to the RMSD simulation illustrated in (Figure 5a). Equilibration was achieved at roughly 47 ns for the Aβ42–compound **2** complex system (Figure 5b). The Aβ42–compound **3** complex system equilibrated after 51 ns, according to the RMSD simulation (Figure 5c). The RMSD simulations, on the other hand, revealed that the Aβ42–compound **4** bound system attained equilibration late during the last 22 ns (Figure 5d), which is consistent with compound **4**’s low activity (aggregation inhibition rate 19%). 

A root mean square fluctuation (RMSF) study can be used to identify more flexible regions and compare the relative flexibility of various components of a system. We examined the flexibility changes during MD simulations in order to uncover some key trends for each of the protein–binder complex systems in our work. For the average flexibility of each amino acid residue in the Aβ42 peptide, the RMSF at the C-α of each residue was taken into account for each peptide–ligand combination of compounds **1**–**4**. For each protein–ligand complex system, the flexibility of the C-α amino acid residue along the Aβ42 sequence was compared to the flexibility of the C-α amino acid residue along the Aβ42 sequence for each of the other complex systems (Appendix A). The RMSF was plotted during the last 10 ns. The presence of peaks indicates the areas that fluctuate most during the simulation. For compounds **1**–**4**, the most fluctuating amino acids residues were Lys16, Leu17, Ala20, Ala21, Val24, Gly25, Ser26, Ala30, Leu34, and Val36. The hydrophobic properties of these amino acids’ side chains suggest that hydrophobic interactions play an important role in the free energy of binding of the models. The RMSF values of the residues were 2.5–3.5 nm for compound **1**, 2.5–4.5 for compound **2**, 3.5–5.5 nm for compound **3**, and 3.5–7.5 nm for compound **4**. The overall flexibility pattern of Aβ42–ligand complex systems (Appendix A) can be used to define some common patterns for the localization of the most flexible areas in the peptide chain. In general, these diagrams reveal that the active-site area, which includes the amino acid residues (Glu11, His14, Gln15, Val18, Phe19, Glu22, and Asn27) responsible for the majority of the interactions with compounds **1**–**4**, has a low average flexibility when compared to the rest of the Aβ42. 

Using the final 10 ns frame time, CPPTRAJ hydrogen bond analysis of the MD simulation results showed the hydrogen bonds of the active site residues Gln15, Glu22, and Asp23 to compound **1**, Glu22 and Ser26 to compound **2**, Asp7 to compound **3**, and no hydrogen bonding amino acids to compound **4** (Appendix A). MM/GBSA methods were used to calculate the binding free energies of the ligands **1**–**4** to Aβ42 utilizing MD simulation trajectories during the last 4 ns (Table 2). The MM/PBSA-per-residue decomposition analysis was used to provide insight into the interactions between the binding site and compounds **1**–**4** [36]. The findings revealed that the amino acid residues His14 and Glu22 play a critical role in effective binding interactions with compounds **1**–**4**, as demonstrated in Table 3 by absolute decomposed energy.

Figure 6 Shows the average structure derived from the last 10 ns trajectory files of the MD simulations process for the Aβ42–ligand complex system for compounds **1**–**4**. For the most active compound **1** (Figure 6a; aggregation inhibition rate 98%), a 3-isobutylenyl-2,4-dihydroxyphenyl fragment is anchored horizontally between Gln15 and Glu22 leaving the isobutylene group extended in the hydrophobic region surrounded by the side chains of Val18 and Phe19. The OH-2 of the 3-isobutylenyl-2,4-dihydroxyphenyl fragment formed a hydrogen bond with the NH2 group of Gln15, whereas the OH-4 formed a hydrogen bond with the oxygen of the carboxylate group of Glu22. These two hydrogen bonds keep the phenyl ring plane horizontal between Gln15 and Glu22, allowing the ligand to stay in an extended conformation alongside the binding site. The phenyl ring of p-hydroxyphenyl formed a µ-cation contact with His14, anchoring the phenyl ring in a coplanar position with the phenyl ring of the 3-isobutylenyl-2,4-dihydroxphenyl fragment. These interactions were supposed as the main participants in the strong binding and the highest activity of compound **1** as inhibitor of the Aβ42 aggregation. In compound **2** (Figure 6b; aggregation inhibition rate 90%) the 5-isobutylenyl-2-hydroxy-4-methoxyphenyl showed a hydrogen bond between the hydroxyl group and Glu22 and Ser26. For compound **3** (Figure 6c; aggregation inhibition rate 68%), the hydroxyl group of p-hydroxyphenyl showed a hydrogen bond with the carboxylate group of Asp7. The representative average structure of compound **4** (Figure 6d; aggregation inhibition rate 19%) showed no hydrogen bonds to the Aβ42 amino acid residues.

Within the neurotoxic fibrils, the Aβ42 peptide structure has a β-turn folded conformation. A salt bridge link between Asp23 and Lys28, as well as hydrophobic interactions between Phe19 and Ile32/Leu34/Val36, His13 and Val40, and Gln15 and Val36, stabilize the β-turn conformation. Side chains compact as a result of these interactions [7,8,9,10]. The binders were stabilized inside the central region of the Aβ42 complexes with compounds **1**–**4** by interactions with the residues Glu11, His14, Gln15, Leu17, Val18, Phe19, Glu22, and Asn27, according to our MD simulation research (Table 3). These interactions inhibit Aβ42 from forming a β-turn fold by disrupting the intramolecular hydrophobic interaction and preventing side chain compacting, which prevents the production of neurotoxic Aβ42 oligomers. Interestingly, the binding free energy ΔG_MM/GBSA_ of the compounds (Table 2: 1, –23.4525; 2, −19.0388; 3, −15.3399; 4, −14.0815 kcal mol^−^^1^) was in accordance with the experimental Aβ42 aggregation inhibitory rate percent (98, 90, 68, and 19%, respectively). Electrostatic energy was the primary contribution to the binders 1–3 and a small contributor to the least active molecule, binder 4, when the components in the MM/GBSA binding free energies were examined. The bound ligands’ strong interactions with His14, Gln15, Val18, Phe19, and Glu22 (Table 3) are thought to have a role in Aβ42 aggregation. The findings of our research on the binding interactions between compounds **1**–**4** and Aβ42 could be relevant in future research, such as the development of more effective Aβ42 aggregation inhibitors. These results agree well with previous published MD simulations of β-amyloid peptides [37,38,39,40], thereby offering promising leads towards development of potential therapeutics for AD.

## 3. Methods

### 3.1. Docking Study

All the 3D structures of compounds **1**–**4** were downloaded from the website of PubChem (Pubchem CID 5281255, 5321765, 10337211, and 14236566, respectively) (Figure 3). These compounds were reported as inhibitors of the aggregation of Aβ42 peptides (Inhibitor rates % are 98, 90, 68, and 19, respectively) [29]. The Aβ42 monomer 3D NMR structure (PDB ID: 1Z0Q) was downloaded from the Protein Data Bank (PDB; www.rcsb.org; accessed on 6 January 2021). Docking studies were performed using Autodock 4.0 [30]. The protein structures were imported and prepared with the default parameters. The whole surface of the Aβ42 monomer was utilized during the docking process. The docking parameters were set to 100 docking runs, 150 populations in the genetic algorithm, and 100 docked poses clustered into groups with root mean square deviations (RMSD) lower than 1.0 Å. The lowest energy pose on the cluster was selected as the binding conformation and was considered for the next MD simulations study.

### 3.2. MD Simulations

The docked structure of 1Z0Q with each of compounds **1**–**4** served as a starting structure for MD simulations using the AMBER 18 program [41] under the Ubuntu 14.2 operating system. The ligand force field parameters were generated using the antechamber module [42], and the topology and parameter files were created using the tleap module. To achieve a neutral solution, the protein–binder complex was intentionally solvated with water molecules while complementary hydrogen atoms and sodium counter-ions were added. The complex structure was encompassed with a 10 Å-thick water-box that was made utilizing the TIP3P water model. The MD simulations were carried out using the AMBER 18 program with a periodic boundary condition and the ff14SB force field. The particle–mesh Ewald protocol was assigned for long-range electrostatic interactions [43] and a 8 Å cut-off value was assigned for non-bonding interactions. At first, 1000-cycle minimizations were used for minimizing water molecules and counter ions followed by 1000-cycle minimizations of the whole system. Then, MD simulations at 298.15 K were allowed to execute on fixed protein coordinates, and equilibrations of a time period of 170 ps were obtained under constant pressure (NPT) and constant normal temperature. In the first 20 ps, the solutes were restrained, while the water molecules and counter ions were equilibrated. The relaxation of the amino acid side chains took the next 50 ps, and all constraints were removed in the last 100 ps. Lastly, 100 ns MD simulations were performed at a temperature of 298.15 K and pressure of 1 atm with a two femtoseconds (fs) time step. The SHAKE algorithm [44] was used to restrain covalent bonds with hydrogen atoms, while Langevin dynamics were employed to control the system’s temperature. The atom coordinates of the system were stored every 10 ps during the MD simulations. The convergence of the simulation processes was validated through the calculation of the RMSDs from the initial structures using the CPPTRAJ module. The root mean square fluctuations (RMSFs) were calculated using the average structures as the reference structures and the final 10 ns of equilibration MD simulations to guess the protein structure’s flexibilities. Throughout the final 10 nanoseconds, the CPPTRAJ module was utilized to examine hydrogen bonding between the binder and the surrounding amino acid residues.

### 3.3. Binding Free Energy Calculation

The binding free energy of snapshots of the MD trajectory were calculated using molecular mechanics/generalized born surface area scoring MM/GBSA [45] for each molecular species (protein, ligand, and protein–ligand complex). Equation (1) was used to compute the binding free energy. The molecular mechanics energy (ΔGMM) was composed of van der Waals and electrostatic interactions. Solvation free energy (ΔGsol) includes the non-polar and the polar contributions. The solvent-accessible surface area (SASA) model is the determinant of the non-polar solvation free energy. Polar solvation free energy was computed by solving the generalized born equation for the MM/GBSA method. Furthermore, the binding free energies were decomposed to a single residue using the molecular mechanic Poisson–Boltzmann surface area MM/PBSA method [46].
ΔG-binding = G-complex − (G-protein + G-ligand)(1)
ΔG = ΔG gas + ΔG solv − TΔS(2)
ΔG gas = ΔE electrostatic + ΔE vdW(3)
ΔG solv = ΔGGB + ΔGSA(4)
ΔGSA = γ × SASA + b(5)

G-complex, G-protein, and G-ligand represent the free energies of the complex, protein, and ligand; ΔG gas represents the gas-phase free energy, comprising electrostatic (ΔE electrostatic) and van der Waals (ΔE vdW) terms; ΔG solv is the solvation energy that comprises polar (ΔGGB) and nonpolar (ΔGSA) contributions. ΔGGB was estimated using the modified GB model informed by Onufriev et al. [47] and εw = 80 and εp = 1.0, and the solvent-accessible surface area (SASA) was calculated using linear combinations of pairwise overlaps [48]. The surface tension proportionality constant (γ) was set to 0.0072 kcal mol^−1^Å^−2^, whereas the nonpolar solvation free energy for a point solute (b) was set to 0.00 kcal mol^−1^.

### 3.4. Clustering Analysis

Clustering analysis [49] is a means of extracting a representative structure from the simulation data through grouping similar conformations together. Distance metric is a similarity measure. Coordinate root mean square deviation (RMSD) is one commonly used distance metric. K-means clustering algorithm was used in the CPPTRAJ cluster command. In our work, it read all the frames of the last 10 ns trajectory file and skipped every 10th frame. The process runs until reaching a state in which there is no longer change clusters of 500 iterations. The RMSD of all atom types except hydrogen was used as a distance metric. Finally, the average over all frames in each cluster was written into files.

## 4. Conclusions

Through its substantial inhibitory effects on Aβ42 aggregation, the anti-Alzheimer effects of four natural compounds that are active ingredients of Psoralea Fructus (PF) were observed, both in vitro and in vivo. The free energy of binding of the compounds is critical for evaluating their anti-aggregation properties and studying the ligand–protein binding mechanism. His14, Gln15, Val18, Phe19, and Glu22 are important for interactions with the ligands, and may be important in the Aβ1-42 aggregation process. These findings could pave the way for future research towards the development of potentially effective anti-AD drugs.

## Figures and Tables

**Figure 1 ijms-23-03546-f001:**
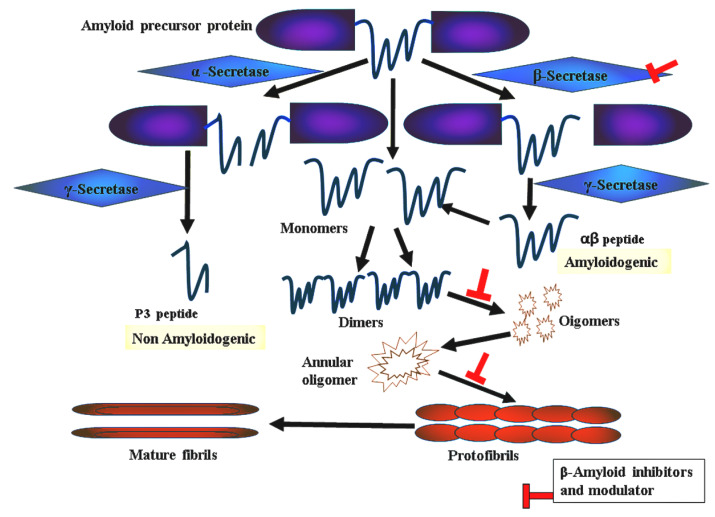
Schematic representation of amyloid plaque formation.

**Figure 2 ijms-23-03546-f002:**
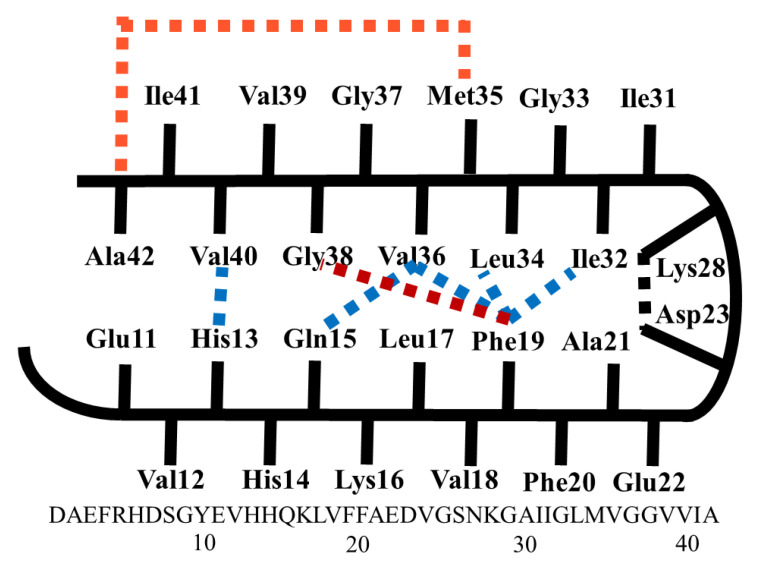
Schematic representation of the monomer structure of Aβ40 and Aβ42 fibrils.

**Figure 3 ijms-23-03546-f003:**
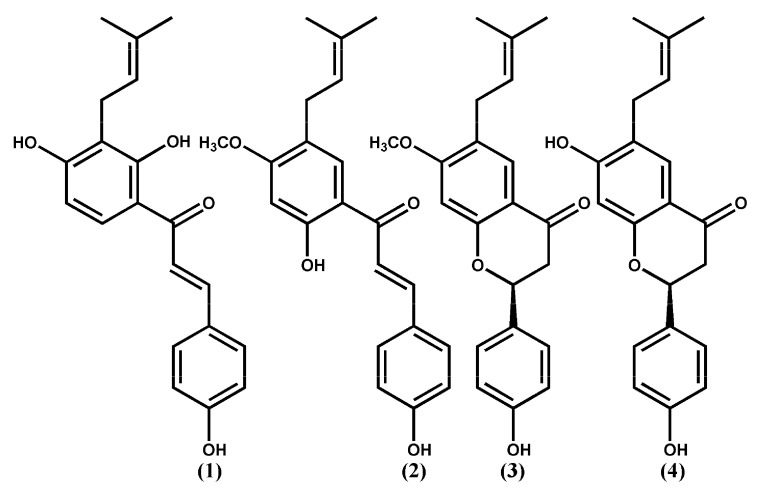
Chemical structures of compounds **1**–**4**.

**Figure 4 ijms-23-03546-f004:**
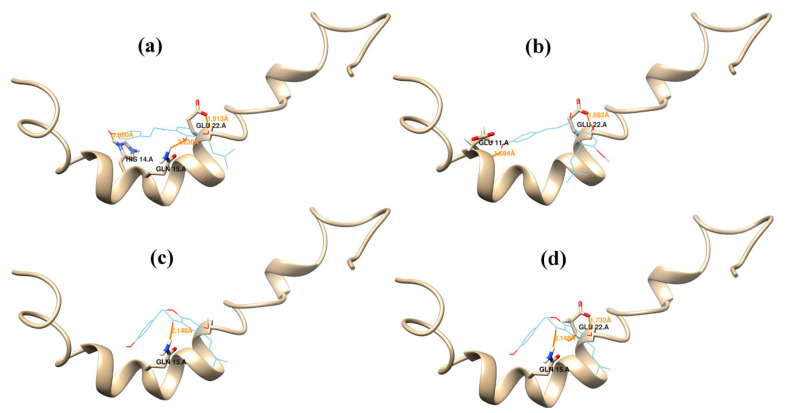
Docking results. Binding conformations of compounds **1** (**a**), **2** (**b**), **3** (**c**), and **4** (**d**) at the binding site of Aβ42. Hydrogen bonds are represented as yellow lines.

**Figure 5 ijms-23-03546-f005:**
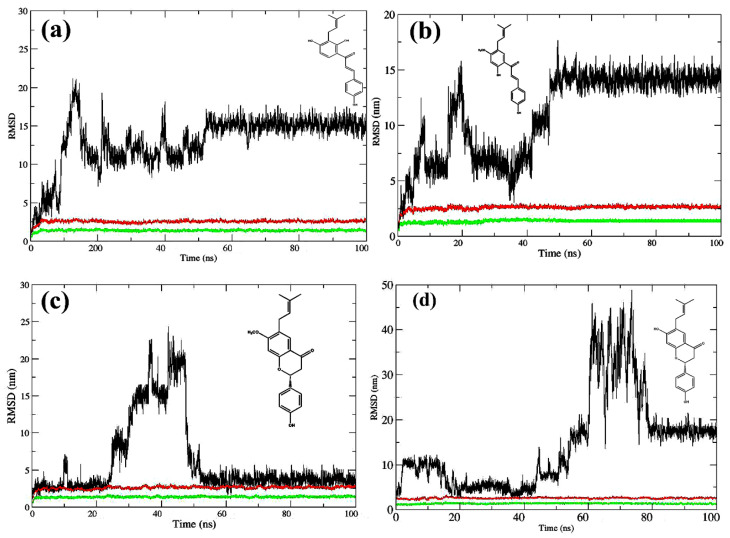
Root mean square deviation (RMSD) in molecular dynamics (MD) simulations of Aβ42 peptide complex with compound **1** (**a**), compound **2** (**b**), compound **3** (**c**) or compound **4** (**d**). RMSD of protein backbones (green lines), proteins with all atoms (red lines), and ligands (black lines).

**Figure 6 ijms-23-03546-f006:**
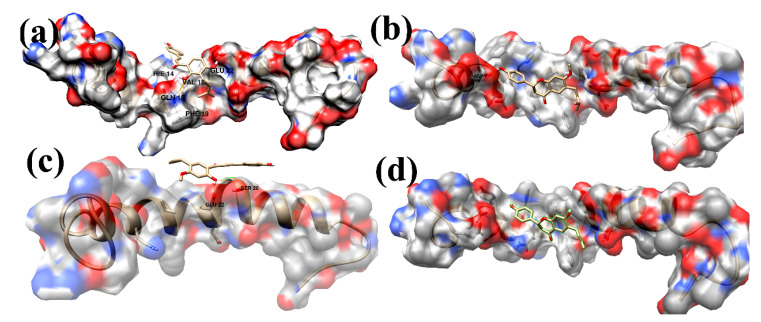
The representative complex structure of AB42-binders as determined by the averaged structure from the last 10 ns MD simulation. The hydrogen bonds are shown in orange solid lines. Compounds **1**–**4**, (**a**–**d**), are shown in a stick representation. The interacting residues and compounds **1**–**4** are shown in a stick representation.

**Table 1 ijms-23-03546-t001:** Amino acid residues of Aβ42 that interact with compounds **1**–**4** and their ΔG values obtained via docking studies.

Compounds	∆G (Kcal Mol^−1^)	Estimated Ki (µM)	Inhibitory Rate % on Aβ142 Aggregations	H.b a.a	VdW a.a
1	−5.23	147.06	98	His14, Gln15, Glu22	Glu11, Val18, Phe19
2	−4.78	315.52	90	Glu11, Glu22	His14, Gln15, Val18, Phe19
3	−4.27	747.57	68	Gln15	Glu11, His14, Val18, Phe19, Phe29, Glu22, Asp23, Asn27
4	−4.00	147.06	19	Gln15, Glu22	Glu11, His14, Val18, Phe19, Asp23

**Table 2 ijms-23-03546-t002:** Calculated binding free energies (kcal mol^−1 a^).

Compounds	ΔGvdw	ΔGelec	ΔGpolarb	ΔGSurfc	ΔGMMGBSA
**1**	−23.8391	−36.6043	40.9619	−3.9710	−23.4525
**2**	−10.5171	−45.8234	40.6926	−3.4022	−19.0388
**3**	−19.8804	−16.0076	23.9728	−3.4274	−15.3399
**4**	−19.0178	−1.8406	8.9947	−2.2198	−14.0815

^a^ Average of 1000 frames; Whole electrostatic contribution: ΔGelec = ΔGelectrostatic + Δgpolar; Whole nonpolar contribution: ΔGnp = ΔGvdw + Δgsurf.

**Table 3 ijms-23-03546-t003:** MM/PBSA-pairwise decomposition analysis of the binding site residue interaction energies to the ligands (kcal mol^−1^).

Aβ42 Residues	Glu11	His14	Gln15	Leu17	Val18	Phe19	Glu22	Asn27
**1**	−0.099	−2.661	−0.896	−0.137	−3.954	−1.879	−5.157	−3.072
**2**	−0.103	−3.669	−2.055	−2.568	−3.754	−0.099	−8.321	−0.403
**3**	−3.233	v1.306	−2.790	−0.166	−4.619	−3.206	−2.569	−3.619
**4**	−5.109	−0.740	−7.221	−0.174	−2.243	−3.324	−3.446	−1.369

## Data Availability

The data are contained within the article and Appendix A.

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
