# Peer review of "Combined Modeling Study of the Binding Characteristics of Natural Compounds, Derived from Psoralea Fruits, to β-Amyloid Peptide Monomer"

_ijms, 2022, doi:10.3390/ijms23073546_

Round 1
Reviewer 1 Report
Authors have studied the interaction of four compounds from Psoralea Fructus with abeta (1-42) monomer using molecular docking and MM-GBSA based binding free energy calculations. The MM-GBSA calculations were based on the molecular dynamics calculations. Authors show that the four compounds show significant binding affinity for the amyloid beta monomer. However, this is not proving the Anti-Alzheimer's property of the compounds. In order to establish this it should be shown that the compounds are able to inhibit the fibril growth or they are able to bust the already formed fibrils. Just by showing that these compounds show binding affinity for monomer of abeta peptide, the anti-Alzheimer's property cannot be claimed. The authors need to carry out simulations to study the fibril busting properties by using amyloid protofibril or fibril models. At this stage I cannot recommend the article for publication in IJMS. The results also need to be compared with that of the known amyloid inhibitors or amyloid buster molecules.
Author Response
Dear Professor
I’m so thankful for your valuable comments and suggestions.
Could you please receive the answer to your comments and suggestions.
Comments and Suggestions for Authors: ………….It should be shown that the compounds are able to inhibit the fibril growth or they are able to bust the already formed fibrils. Just by showing that these compounds show binding affinity for monomer of abeta peptide, the anti-Alzheimer's property cannot be claimed. The authors need to carry out simulations to study the fibril busting properties by using amyloid protofibril or fibril models. At this stage I cannot recommend the article for publication in IJMS. The results also need to be compared with that of the known amyloid inhibitors or amyloid buster molecules.
>> Answer:
There several in silico studies of the binding affinity of the beta amyloid peptides to small molecules with potential anti-Alzheimer’s activity. Several studies were based on peptide monomers [1-3] and other studies were based on fibril models.
Within the neurotoxic fibrils, the Aβ42 peptide structure has a β-turn folded conformation. A salt bridge link between Asp23 and Lys28, as well as hydrophobic interactions between Phe19 and Ile32/Leu34/Val36, His13 and Val40, and Gln15 and Val36, stabilize the β-turn conformation. Side chains compact as a result of these interactions [4-7]. In our work, we study the interaction of compounds 1-4 with these amino acid residues. These interactions inhibit Aβ42 from forming a β-turn fold by disrupting the intramolecular hydrophobic interaction and preventing side chain compacting, which prevents the production of neurotoxic Aβ42 oligomers. The binding free energy ΔGMM/GBSA of the complexes also help in understanding the pharmacophoric groups needed in the design of new lead molecules.
- Biomacromol. J. Vol. 1, No. 2, 230-241, December 2015.
- Molecules 2018, 23, 614
- Netw Model Anal Health Inform Bioinforma (2013) 2:13–27
- Tycko, R. Q. Rev. Biophys. 2006, 39, 1–55.
- Paravastua, A.K.; Leapman, R.D.; Yau, W.M.; Tycko, R. Proc. Natl. Acad. Sci. USA. 2008, 105, 18349–18354.
- Lührs, T.; Ritter, C.; Adrian, M.; Riek-Loher, D.; Bohrmann, B.; Döbeli, H.; Schubert, D.; Riek, R. Proc Natl Acad Sci USA. 2005, 102, 17342–17347.
- Masuda, Y.; Uemura, S.; Nakanishi, A.; Ohashi, R.; Takegoshi, K.; Shimizu, T.; Shirasawa, T.; Irie, K. Bioorg. Med. Chem. Lett. 2008, 18, 3206–3210.

Reviewer 2 Report
Major comments
In this study, the authors tried to identify the important residues for binding of flavonoid compounds derived from Psoralea Fruits to amyloid-β42 (Aβ42), by in silico analyses. Findings are potentially interesting and important; however, there are issues of serious concerns.
- Results and discussions section;
Although the authors combined Results section with Discussion section, the resultant section substantially included results only; the authors provided explanations of the results obtained, but discussion was superficial. I suggest that the authors discuss in detail
- the potential mechanisms underlying the different inhibitory effects of 4 different compounds on Aβ42 aggregation.
- residues/structures which are important for future studies to focus on, in order to design more effective aggregation inhibitors.
Minor comments
- MD (line 52); abbreviations must be spelled out completely on initial appearance in text.
- S1-S3 (line 100); I think that this is S1-S4.
Author Response
Dear Professor
I’m so thankful for your valuable comments and suggestions.
With track changes, the responses to the comments were typed within the manuscript.
Major comments
- Results and discussions section;
Although the authors combined Results section with Discussion section, the resultant section substantially included results only; the authors provided explanations of the results obtained, but discussion was superficial. I suggest that the authors discuss in detail
- the potential mechanisms underlying the different inhibitory effects of 4 different compounds on Aβ42 aggregation.
- residues/structures which are important for future studies to focus on, in order to design more effective aggregation inhibitors.
>>> Answer: Page 9, 10; Lines 161-175. The manuscript has been updated to include more discussion of the findings.
Minor comments
- MD (line 52); abbreviations must be spelled out completely on initial appearance in text.
- S1-S3 (line 100); I think that this is S1-S4.
>>> Corrections were done.

Round 2
Reviewer 1 Report
I do not have any bad intention against this work and my following comments are based on my research in this area for many years. I am not convinced that the authors are trying to establish the anti-Alzheimer's property of the compounds by showing their interaction with monomer. More over the binding energies cannot be taken literally and many time even non-inhibitor molecules can also show negative binding free energies against these targets. So, it needs to be established that the compounds also bind to protofibrils or fibrils with higher binding affinity similar to other known inhibitors. As of now, from the calculations what is shown is that these compounds bind to monomer of amyloid beta-peptides and in no way this establishes the anti-Alzheimer's property of these compounds. I am unable to recommend the manuscript in its present form for publication in IJMS.
Author Response
Comments and Suggestions for Authors: I do not have any bad intention against this work and my following comments are based on my research in this area for many years. I am not convinced that the authors are trying to establish the anti-Alzheimer's property of the compounds by showing their interaction with monomer. More over the binding energies cannot be taken literally and many time even non-inhibitor molecules can also show negative binding free energies against these targets. So, it needs to be established that the compounds also bind to protofibrils or fibrils with higher binding affinity similar to other known inhibitors. As of now, from the calculations what is shown is that these compounds bind to monomer of amyloid beta-peptides and in no way this establishes the anti-Alzheimer's property of these compounds. I am unable to recommend the manuscript in its present form for publication in IJMS.
>> >> Answer: Our MD simulation study was shedding light on the mechanism of the aggregation inhibitor activity of the compounds under study. In view of the fact that Aβ42 peptide aggregation require the amyloid peptide monomers in its beta-structure, we studied the interfering effect of the that compounds on the side chain and inhibiting the Aβ42 from forming a β-turn fold by disrupting the intramolecular hydrophobic interaction and preventing side chain compacting, which prevents the production of neurotoxic Aβ42 oligomers.
There several in silico studies of the binding affinity of the beta amyloid peptides to small molecules with potential anti-Alzheimer’s activity. Several studies were based on peptide monomers [1-3] and other studies were based on fibril models.
Within the neurotoxic fibrils, the Aβ42 peptide structure has a β-turn folded conformation. A salt bridge link between Asp23 and Lys28, as well as hydrophobic interactions between Phe19 and Ile32/Leu34/Val36, His13 and Val40, and Gln15 and Val36, stabilize the β-turn conformation. Side chains compact as a result of these interactions [4-7]. In our work, we study the interaction of compounds 1-4 with these amino acid residues. These interactions inhibit Aβ42 from forming a β-turn fold by disrupting the intramolecular hydrophobic interaction and preventing side chain compacting, which prevents the production of neurotoxic Aβ42 oligomers. The binding free energy ΔGMM/GBSA of the complexes also help in understanding the pharmacophoric groups needed in the design of new lead molecules.

Reviewer 2 Report
I think that the authors appropriately responded to comments raised in the 1st round of peer-review process, and the revised manuscript has been improved and become more interesting. I have no more suggestions.
Author Response
Comments and Suggestions for Authors: I think that the authors appropriately responded to comments raised in the 1st round of peer-review process, and the revised manuscript has been improved and become more interesting. I have no more suggestions.
Answer: Dear Professor,
I’m so thankful for your valuable comments and suggestions
Round 3
Reviewer 1 Report
Authors have not made any attempt to improve the manuscript based on my suggestions. They are trying to establish the drug like properties of the compounds extracted from Psoralea Fructus by studying their interactions with monomers of amyloid peptides. In order to demonstrate the Anti-Alzheimer's property of the compounds rather the amyloid busting or fibrillation inhibition properties should be shown. So, I cannot recommend the publication of this manuscript in IJMS. Authors should change the Anti-Alzheimer's in the title and they need to correctly mention that they are only studying the interaction of the compounds with the monomers of Amyloid beta peptide.
Author Response
Dear Professor
I'm so thankful for your comments and suggestions.
>> The title of the manuscript was replaced into "Combined Modeling Study of The Binding Characteristics of Natural Compounds, Derived from Psoralea Fruits, to β-Amyloid Peptide Monomer"
All answers are made to the manuscript while Track Changes Feature turned on.
With my great appreciation
Prof. Awwad Radwan
